# A New Approach for a Safe and Reproducible Seeds Positioning for Diffusing Alpha-Emitters Radiation Therapy of Squamous Cell Skin Cancer: A Feasibility Study

**DOI:** 10.3390/cancers14010240

**Published:** 2022-01-04

**Authors:** Giacomo Feliciani, Salvatore Roberto Bellia, Massimo Del Duca, Giorgio Mazzotti, Manuela Monti, Ignazio Stanganelli, Yona Keisari, Itzhak Kelson, Aron Popovtzer, Antonino Romeo, Anna Sarnelli

**Affiliations:** 1Medical Physics Unit, IRCCS Istituto Romagnolo per lo Studio dei Tumori (IRST) “Dino Amadori”, 47014 Meldola, Italy; giorgio.mazzotti@irst.emr.it (G.M.); anna.sarnelli@irst.emr.it (A.S.); 2Radiotherapy Unit, IRCCS Istituto Romagnolo per lo Studio dei Tumori (IRST) “Dino Amadori”, 47014 Meldola, Italy; salvatore.bellia@irst.emr.it (S.R.B.); massimo.delduca@irst.emr.it (M.D.D.); antonino.romeo@irst.emr.it (A.R.); 3Biostatistics and Clinical Experimentation, IRCCS Istituto Romagnolo per lo Studio dei Tumori (IRST) “Dino Amadori”, 47014 Meldola, Italy; manuela.monti@irst.emr.it; 4Skin Cancer Unit, IRCCS Istituto Romagnolo per lo Studio dei Tumori (IRST) “Dino Amadori”, 47014 Meldola, Italy; ignazio.stanganelli@irst.emr.it; 5Department of Clinical Microbiology and Immunology, Tel Aviv University, Tel Aviv 69978, Israel; keisari@post.tau.ac.il (Y.K.); kelson@tauex.tau.ac.il (I.K.); 6Sharett Institute of Oncology, Hadassah Medical Center, Hebrew University of Jerusalem, Jerusalem 91120, Israel; aronpopovtzer@yahoo.com

**Keywords:** squamous cell skin cancer, alpha-emitters radiation therapy, interstitial brachytherapy, treatment safety

## Abstract

**Simple Summary:**

The Diffusing Alpha-emitters Radiation Therapy (DaRT) is a novel brachytherapy technique employing 224-radium enriched seeds releasing short-lived alpha-emitting atoms into the tumour. DaRT overcomes the main obstacle in employing alpha radiation for cancer treatments in liquid and solid media caused by their short range. The aim of the study is to improve the DaRT technique with an external radio-opaque template that can help clinicians predict the correct number of sources to achieve tumour coverage. Furthermore, the template is used to aid clinicians in visualizing lesions and their eventual subcutaneous extension. Finally, it is also utilized on treatment day to ensure that the sources are properly inserted into the tumour.

**Abstract:**

The purpose of this study is to discuss how to use an external radio-opaque template in the Diffusing Alpha-emitters Radiation Therapy (DaRT) technique’s pre-planning and treatment stages. This device would help to determine the proper number of sources for tumour coverage, accounting for subcutaneous invasion and augmenting DaRT safety. The procedure will be carried out in a first phase on a phantom and then applied to a clinical case. A typical DaRT procedure workflow comprises steps like tumour measurements and delineation, source number assessment, and therapy administration. As a first step, an adhesive fiberglass mesh (spaced by 2 mm) tape was applied on the skin of the patient and employed as frame of reference. A physician contoured the lesion and marked the entrance points for the needles with a radio opaque ink marker. According to the radio opaque marks and metabolic uptake the clinical target volume was defined, and with a commercial brachytherapy treatment planning system (TPS) it was possible to simulate and adjust the spatial seeds distribution. After the implant procedure a CT was again performed to check the agreement between simulations and seeds positions. With the procedure described above it was possible to simulate a DaRT procedure on a phantom in order to train physicians and subsequently apply the novel approach on patients, outlining the major issues involved in the technique. The present work innovates and supports DaRT technique for the treatment of cutaneous cancers, improving its efficacy and safety.

## 1. Introduction

The current standard of care for cutaneous squamous cell skin carcinoma (cSCC) is surgical excision, which involves removing the tumour while leaving a margin of healthy skin. If surgery is not an option or is not viable or ineffective due to tumour resistance, gamma or beta-based external beam radiation and brachytherapy can be used as alternatives [1,2,3,4,5,6]. However, these techniques may be ineffective due to hidden tumour invasion or radio resistance [7].

The Diffusing Alpha-emitters Radiation Therapy (DaRT) is a novel brachytherapy technique employing ^224^radium (Ra-224) enriched seeds releasing short-lived alpha-emitting atoms into the tumour [8]. The efficacy of DaRT was proven in a series of preclinical studies on tumours with different histology such as squamous cell, colon, breast, pancreas, lung, and prostate carcinoma [9,10]. Furthermore, tumour abolition by alpha DaRT resulted in activation of specific anti-tumour immunity [11] and an abscopal effect in one patient [12].

This technique improves standard gamma-based brachytherapy treatments with the high relative biological effectiveness of alpha particles and higher control of dose to organs at risk in the surrounding area of the implant [13]. Alpha radiation is generally more effective against tumours, achieving a higher degree of cell killing probability for a given absorbed dose and being insensitive to poor oxygenation of tissues [14]. The main obstacle in employing alpha radiation for cancer treatments is the short range of alpha particles in liquid and solid media (i.e., water and tissue). DaRT overcomes this limitation because Ra-224 deposited on the surface of implantable seeds decays directly into the tumour in ^220^radon and its progeny, which migrate away from the implant site because of recoil energy and its gaseous nature, for an average distance of 2.5 mm [15].

DaRT is for all intents an interstitial brachytherapy technique, an effective and short procedure, and it is clinically employed for the treatment of cutaneous squamous cell carcinoma (cSCC) at our center Istituto Romagnolo per lo Studio dei Tumori “Dino Amadori”—IRST IRCCS (Meldola, FC, Italy). The technique has recently concluded the “first in man” clinical trial N. CTP-SCC-00 (NCT03015883) with promising results showing a complete response tumour rate of 78.6% and a partial response rate of 21.4% of the lesions [16]. The pre-treatment step, which includes the need to predict the correct number of Ra-224 seeds to implant based on tumour dimension and shape days in advance, is a major issue in completing this operation. To date, no commercial treatment planning system exists that can calculate alpha particle dose, and the source estimation is generally performed geometrically on the patient’s skin and/or on a preplanning CT or PET/CT, with the sources spaced about 4–5 mm apart to reach the killing dose in the entire clinical target volume (CTV). We propose and describe how to use an external radio-opaque template in the pre-planning and treatment phases of the DaRT technique to help evaluate the correct number of sources for cSCC. The template is used to aid clinicians in visualizing lesions and their eventual subcutaneous invasion during the pre-planning procedure. It is also utilized on treatment day to ensure that the sources are properly inserted into the tumour and to monitor for tumour progression.

To begin, the treatment will be carried out on a phantom to demonstrate the efficacy of the entire approach as well as the physician training required to safely implant the seeds without the involvement of the patient. Second, the procedure will be demonstrated on a patient to demonstrate its utility in clinical settings.

## 2. Materials and Methods

### 2.1. DaRT Technique Overview

DaRT technique relies on the implantation of 1-cm long Ra-224 loaded seeds directly in the tumour site through needle applicators, as shown in Figure 1. After a little training on the specific applicator, a physician confident with interstitial brachytherapy or surgery should be able to perform the treatment with ease. Except for implantation near major blood vessels, where a safety margin of 1 cm should be used, the operation has no specific contraindications.

Each seed is loaded with an average activity of 2 µCi and can be ordered in series up to a length of 6 cm. A standard DaRT technique workflow is presented in Figure 2, which includes phases such as tumour measurements and delineation, source number, evaluation, and treatment delivery. In the pre-treatment phase, the radiation oncologist draws the CTV directly on the patient’s skin with a surgical pen and marks the needle insertion locations with a ruler, as shown in Figure 2a, considering 4–6 mm margins from the visible gross tumour volume (GTV) to account for microscopic tumour invasion. The inter-distance between insertion points is approximately 4–5 mm to assure a tumour dose coverage of at least 10 Gy. The patients subsequently undergo a CT or PET/CT scan to check lesion subcutaneous extension (Figure 2b). Another seed number estimation is accomplished geometrically on a commercial TPS for traditional brachytherapy in collaboration with a medical physicist (Figure 2c). After sources arrival, seed implantation is performed manually by the radiation oncologist (Figure 2d). A CT scan after implant is performed to check for correct displacement of the sources. At the end, the tumour coverage is ensured a-posteriori by either follow up in tumour complete response cases or with biopsy in suspected ones.

Given the novelty of the technique we refer to the work by Popovtzer et al. [13] and Arazi et al. [12] for further details about seeds insertion technique and dosimetric aspects of 224-Ra.

### 2.2. Template Based Planning Phase 1: Phantom Training

Figure 3 shows a phantom (turkey leg) that replicates a human skin surface. In Figure 3a, a brachytherapist draws a superficial rectangular lesion with a surgical marker that simulates a squamous skin cancer visible on the phantom’s surface, as well as five needle insertion and exit points spaced approximately 4–5 mm and displaced on two planes to simulate a subcutaneous lesion invasion of approximately 6 mm (two 224-Ra loaded seeds disposed on planes one over the other can safely cover a maximum of 8 mm depth with tumour killing dose of 10 Gy in case of subcutaneous tumour invasion). The lesion and its surroundings are covered with a sterile transparent patch, which is then covered with an adhesive fiberglass mesh tape (Figure 3b). To achieve a smooth surface, a layer of silk adhesive tape was applied to the mesh. The physician uses a silver-based non-toxic radio-opaque ink to copy the contouring done on the skin on this surface. Finally, three radio-opaque spots are drawn on the fiberglass mesh and then tattooed onto the skin with black ink in order to reapply the tape on implant day, as indicated in Figure 3b by the red points.

A pre implant CT is then performed where the radio opaque trace will be clearly visible together with the insertion points. Using the commercial treatment planning system (TPS) Oncentra Brachy, Elekta, Stockholm, Sweden, contouring of the hypothetical lesion CTV is performed on CT by following the radio-opaque trace and extending it to a depth of 6 mm to simulate a deep lesion. Geometrical planning is finally performed using the TPS software following the radio opaque insertion points. Planned seeds are implanted and the phantom is scanned again on CT. Pre and post implant CT are then fused for coverage verification through MIMmaestro imaging suite (MIM Software Inc., Beachwood, OH, USA). The fusion process is performed employing a bounding box around the lesion and its radio-opaque trace.

### 2.3. Template Based Planning Phase 2: Patient Application

The technique outlined in the previous paragraph and represented in Figure 3 is now executed on the patient. The patient is first visited by a brachytherapist, who draws the lesion with a surgical marker. After that, a sterile patch was placed, followed by fiberglass mesh. On the day of the intervention, the lesion drawing is duplicated on the fiberglass mesh with radio-opaque ink, and tattoos are applied to the skin as a frame of reference to reposition the mesh tape. At this time, a PET/CT or CT scan is performed to determine the extent of the tumour’s subcutaneous extension, which is done by contouring the gross tumour volume (GTV) on the TPS and expanding it by 4–6 mm to define the CTV according to the NCCN and GEC-ESTRO guidelines [1,17]. The template is adjusted if there is a mismatch with the original drawing. The new template is reapplied to the lesion on the day of the intervention, aligning the radiopaque sites with the tattoos. This allows the physician to mark the correct extension of the CTV on the skin and adequately cover it with planned sources.

## 3. Results

### 3.1. Template Based Planning Phase 1: Phantom Training

Following the phantom preparation described in the template-based planning: phantom training section and displayed in Figure 3, the CT image of the phantom is presented in Figure 4, highlighting the visibility of the fiberglass mesh and the lesion drawn with silver-based ink. The needle applicator’s entrance and exit locations are visible near the lesion. On the TPS needle, applicators are represented as catheters for the conventional brachytherapy planning, whereas 224-Ra seeds extremities are simulated by active source points spaced of 1 cm, as depicted in Figure 5a. Geometrical arrangement of the seeds is done keeping 2 mm from the surface of the skin and a distance between seeds of a maximum of 5 mm, as shown in Figure 5a,b. The distribution of needles and seeds required to properly cover the target from a superficial point of view (coronal view) is shown in Figure 5a, whereas the axial view of the phantom and the need for a second plane below the first to completely cover the red contoured target area is shown in Figure 5b.

In Figure 5c needles are inserted according to the drawing and the phantom is scanned again and fused with the planning CT. Fusion is performed aligning radio opaque marker as can be seen in the animated GIF in Appendix A. In Figure 6, a detail of the first and second seed layer is shown. The first seed layer is measured to be at 2.6 mm from the surface, whereas the second layer is at 6.0 mm from the surface and 3.4 mm from the first layer.

### 3.2. Template Based Planning Phase 2: Patient Application

As shown in Figure 7, the previously noted technique was performed on the patient. The contouring of the patient lesion GTV in Figure 7a is depicted in green, with its 6 mm isotropic expansion to account for tumour margins and microscopic invasion shown in blue denoting the CTV.

The mesh tape was placed upon the patch and the skin-drawn lesion reported with the radiopaque ink on the mesh, as shown in Figure 7b. Localization point marks were placed at the corner of the tape and CT scan was performed. In Figure 8a, the CTV drawn directly on the skin by the physician is shown in blue and subsequently, in Figure 8b, three seeds are planned in order to cover the blue target.

The physician now outlines a second CTV in red based on the information from the PET/CT scan about the tumour’s subcutaneous extension. The discrepancy between the visible CTV and the PET/CT based CTV is greater than 6 mm, as shown Figure 8c, resulting in a change in the number of seeds required to completely cover the target from three to four. The CTV drawn on the skin, even accounting for large margins (5–6 mm), does not consider subcutaneous invasion nor possible tumour growth that may led to dangerous target miss and possible recurrence.

As a result of these considerations, the required number of sources and seed displacement were updated evaluating the amount of space to fully cover the PET/CT based CTV directly on the radio-opaque mesh, as shown in Figure 8d. In this case, an expansion of the CTV of 6 mm is required (correspondent to three steps on the radio-opaque mesh). The physical template is then updated according to the previous considerations and displayed in Appendix A.

## 4. Discussion

The DaRT technology recently completed the “first in man” clinical study N. CTP-SCC-00 (NCT03015883), and the Food and Drug Administration of the United States awarded it the “Breakthrough Device Designation”. As previously stated by Arazi et al. [8], this technique is based on the direct insertion of 224-Ra loaded sources into the tumour.

Alpha particles are extremely effective in destroying cancer cells in comparison to standard photon-based techniques, however, their limited range in tissue represent a strong limitation for their application in clinical setting. A requirement to employ particle based radiotherapy is great precision in target localization and dose delivery. An advantage of this technique is that alpha particles are deposited directly into the tumour without the need of large and very expensive technological facilities to accelerate hadrons such for particle external beam radiotherapy. However, precision in target coverage is still a central issue for the DaRT technique. We demonstrated step by step the feasibility of a template based procedure in a phantom and subsequently applied it to a patient, leading to several major benefits in treatment precision and safety.

The direct visualization of the lesion on CT via the radio-opaque template will assist physicians and medical physicists in better evaluating lesion extension and, as a result, will provide a better estimation of the number of sources required to achieve full tumour coverage while minimizing needle insertions, as shown in Figure 8.

Furthermore, by comparing the naked eye and PET/CT based lesion CTV, millimetric sized subcutaneous lesion extensions that are very difficult to spot can be analyzed. Even increasing the naked eye-based GTV by 6 mm to account for tumour subcutaneous extension and microscopic invasion, the lesion is not totally encompassed by the CTV for another 6 mm, as shown in Figure 8d. This difference may appear irrelevant for photon based techniques, but it becomes crucial when dealing with alpha particles where a maximum distance of 2.5 mm is tolerated to assure 10 Gy tumour killing dose coverage. Without template guidance in this case, it is very likely that a target miss will occur during the implant, with probable tumour relapse as a result.

In Figure 6 we can observe image fusion between pre and post implant CT on a phantom where we tried to force radio-opaque ink marks to match. Although we were able to evaluate the intervention results with sub millimetric precision using this comparison, we were only able to achieve fair results using standard fusion approaches due to lesion swell generated by the insertion of the sources, as seen in Appendix A. This is the study’s primary limitation, as the swell will be significantly more prominent in patients. Advanced fusion approaches, such as elastic registrations, can be used to mitigate this problem, but this investigation is beyond the scope of this work. Regardless of this difficulty, it is still possible to determine that seeds were planted and spaced at a maximum of 4 mm apart. Another limitation of this study is that we cannot provide complete lesion margins assessment with peripheral and deep en face margin assessment (PDEMA) technique, but we have to rely on the indications provided by NCCN and GEC-ESTRO guidelines for considerations about target definition and treatment for conventional radiotherapy.

The procedure’s flexibility in considering tumour subcutaneous extension and potential growth is the procedure’s last benefit. It is possible to quantify the adjustments required for tumour coverage using the TPS and, as a result, update the template. New tumour contours and needle insertion locations coordinates are derived from the radio-opaque mesh and used to update the physical template, eliminating target misses and optimizing seeds positioning, as shown in Figure 8d and Appendix A.

## 5. Conclusions

The DaRT’s safety and precision were improved in this study, which included a detailed description of physician training on phantom and clinical applications. In particular, when dealing with subcutaneous extension of the tumour, the use of a template may avoid dangerous target miss and ensure complete dose coverage.

## Figures and Tables

**Figure 1 cancers-14-00240-f001:**
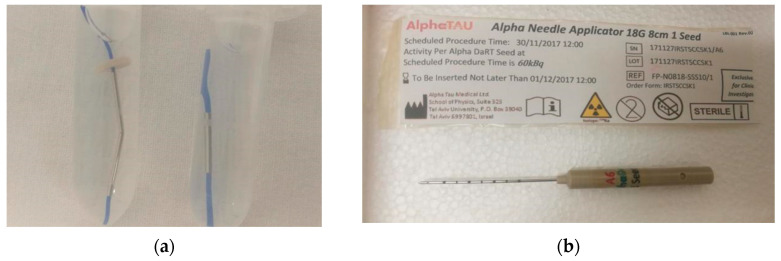
Image of 2-cm and 1-cm long DaRT 224-Ra source seeds (**a**) and needle applicator for seeds insertion (**b**).

**Figure 2 cancers-14-00240-f002:**
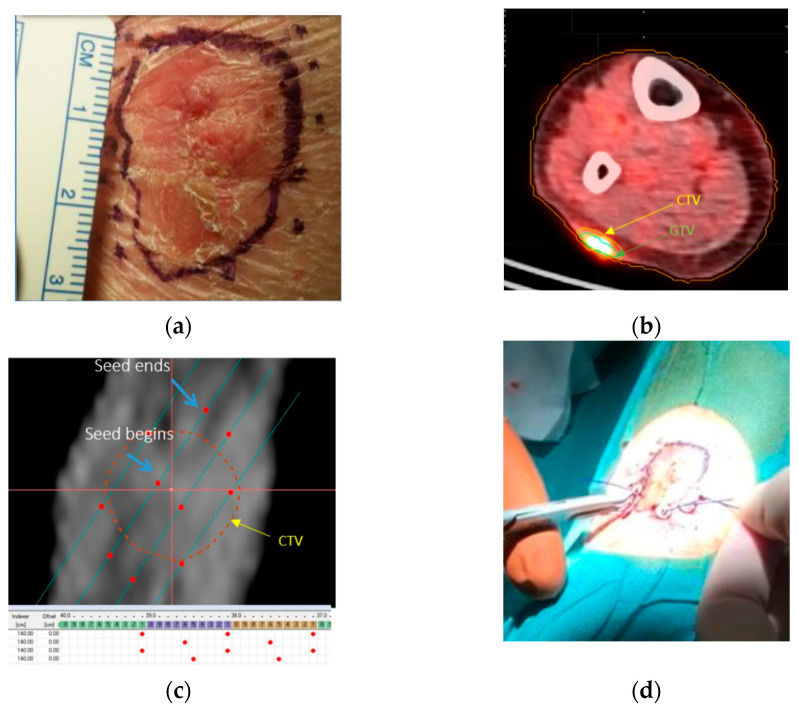
(**a**) Lesion is contoured by the physician directly on the skin of the patient and insertion points are theoretically marked with surgical blue ink. (**b**) Gross tumour volume (GTV) of the lesion is contoured on a PET/CT scan and clinical tumour volume is obtained as expansion of 4–6 mm of the GTV in order to account for microscopic tumour invasion and eventual patient swell during intervention. (**c**) Commercial TPS is employed to geometrically evaluate the number of seeds needed for full tumour coverage. (**d**) Seed implantation is performed by the radiotherapist.

**Figure 3 cancers-14-00240-f003:**
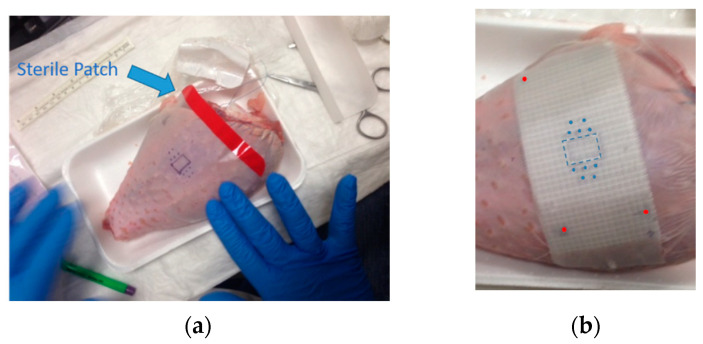
(**a**) Phantom lesion is contoured by the physician with surgical blue ink and a transparent sterile patch is superimposed (indicated in figure by the blue arrow). (**b**) Adhesive fiberglass mesh is applied over the patch in order to re-draw the lesion with radio-opaque ink (marked in blue) and the localization points used as reference points (marked with red dots).

**Figure 4 cancers-14-00240-f004:**
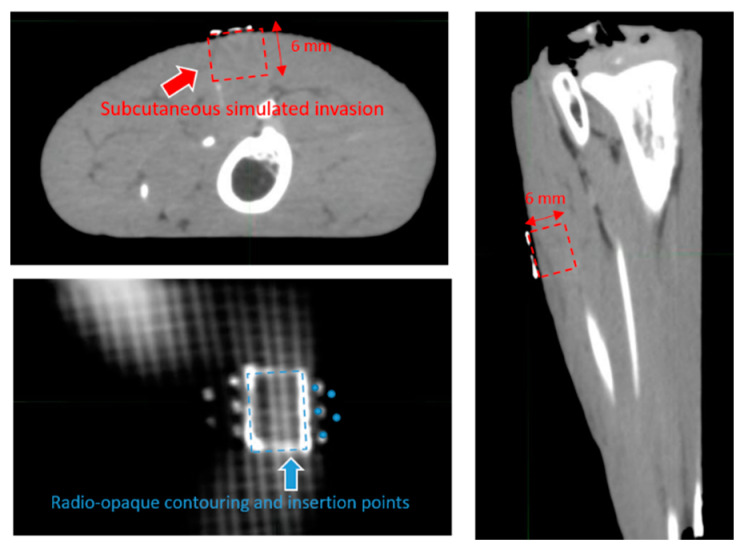
Details of the visibility of the radio-opaque ink in axial coronal and sagittal plane.

**Figure 5 cancers-14-00240-f005:**
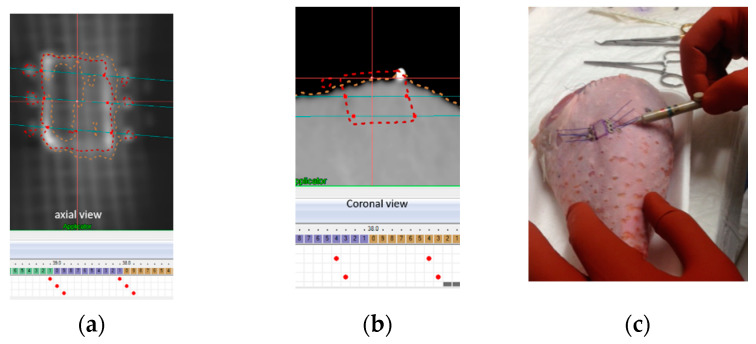
(**a**) CT coronal view of the radio-opaque contouring of the lesion with needles’ insertion and exit points. (**b**) Axial view of the lesion contoured with dotted red line and brachytherapy catheters reconstruction simulating needles displacement. (**c**) Detail of the needle insertion procedure.

**Figure 6 cancers-14-00240-f006:**
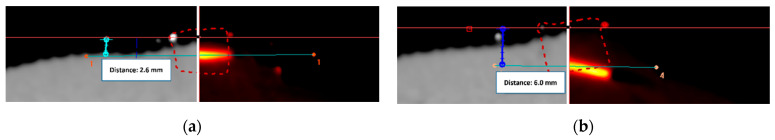
Pre- and post-implant phantom CT fusion, showing the first (**a**) seed insertion plane at 0.26 cm from the surface and the second (**b**) insertion plane at 0.6 cm from the surface.

**Figure 7 cancers-14-00240-f007:**
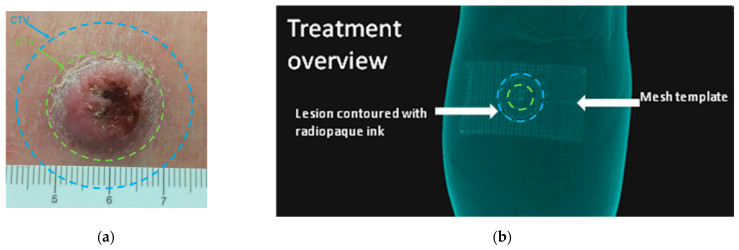
(**a**) GTV contouring is shown in green, with its enlargement to a 6 mm CTV shown in blue. (**b**) Three-dmensional rendering of the radio-opaque template with details of lesion GTV, CTV, and mesh template.

**Figure 8 cancers-14-00240-f008:**
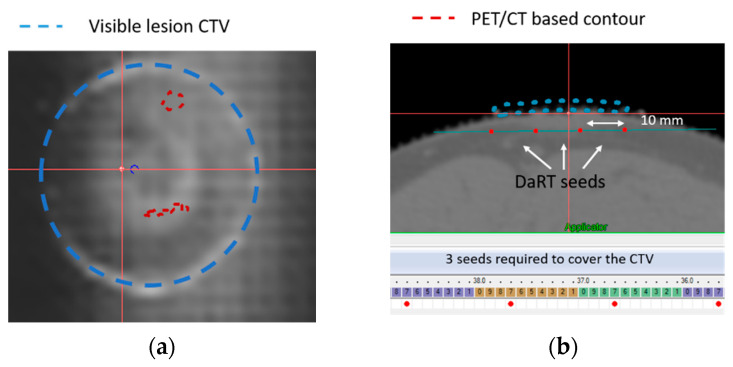
(**a**) Visible CTV drawn with the radio-opaque marker by the physician highlighted in blue over the radio-opaque trace. (**b**) Geometrical planning on CT with the TPS with the 1 cm long DaRT seeds represented by red dots. The coverage of blue CTV is achieved with three seeds. (**c**) The PET/CT-based CTV is depicted in red alongside the blue CTV, indicating subcutaneous tumour invasion and the necessity for four DaRT seeds for proper lesion coverage. (**d**) Final estimation of the CTV is drawn on the template mesh in order to update the physical template.

## Data Availability

The data presented in this study are available in this article (and Appendix A).

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
