# Peer review of "A New Approach for a Safe and Reproducible Seeds Positioning for Diffusing Alpha-Emitters Radiation Therapy of Squamous Cell Skin Cancer: A Feasibility Study"

_cancers, 2022, doi:10.3390/cancers14010240_

Round 1

Reviewer 1 Report

Title

A new approach for a safe and reproducible seeds positioning for Diffusing Alpha-emitters Radiation Therapy of squamous cell skin cancer: a feasibility study.

Concise Summary

The authors describe a new procedure to administer a relatively new type of radiotherapy based in alpha-particles for the treatment of cutaneous squamous cell carcinoma (cSCC). By this procedure, the authors propose an improving in the efficacy and safety of the implant. The clinical target volume and the spatial seeds distribution are obtained through a radio opaque marks and metabolic uptake based on Positron Emission Tomography (PET) and CT scan. Finally, the authors conclude it is possible to administer alpha-particles in an efficient way according to the proposed protocolized procedure.

Questions

The authors aim that the proposed technique relies in the fact that alpha particles are deposited directly into the tumor without the need of large and very expensive technological facilities. However, this procedure requires of PET/CT scan. Could the authors explain why they consider the procedure to be easy to apply?

There is not enough high-quality evidence about the effectivity of radiotherapy in the management of cSCC. What clinical studies support that the alpha-particle administration procedure explained in the article is sufficient to obtain tumor coverage in cSCC?

How it is possible to detect microscopic subcutaneous expansions by the procedure proposed by the authors? Could the authors give enough data to support this statement?

Although surgical excision is considered to be the primary treatment approach for curative treatment of cSCC, radiotherapy can play a role in both the definitive and adjuvant settings. Which results support that the alpha-particle administration is efficient in treating the real lesion borders in cSCC?

Author Response

First of all, we thank the Reviewer whose comments helped us to improve the manuscript.

Hereafter, “C# stands for Reviewer’s comments and “R#” for our replies. All the reference numbers (pages, citations, etc.) refer to the “new” version of the text, unless explicitly otherwise defined. In the answers, the references to manuscript’ sentences are reported between “< … >”. All changes to the text are highlighted with Word Revision tool or reported in red.

COMMENT OF REVIEWER#1:

[C1.1]: The authors aim that the proposed technique relies in the fact that alpha particles are deposited directly into the tumor without the need of large and very expensive technological facilities. However, this procedure requires of PET/CT scan. Could the authors explain why they consider the procedure to be easy to apply?

[R1.1]: We thank the Reviewer for the comment. The procedure requires a CT scanner like any other radiotherapy technique and a PET/CT is only recommended and not necessary. In the introduction we specify in line 74 that “ the source estimation is generally performed geometrically on the patient's skin and/or on a preplanning CT or PET/CT, with the sources spaced about 4-5 mm apart to reach the killing dose in the entire Clinical Target Volume (CTV)” and in the methods section line 108 “The patients subsequently undergo a CT or PET/CT scan to check lesion subcutaneous extension”.

We argue that the procedure is easy to apply because we consider CT availability as a standard in a cancer center or in a hospital in general. Furthermore, after a brief training for the physicians confident with interstitial brachytherapy performing the needle insertion and for medical physicists performing seeds estimations the procedure becomes quite straightforward in comparison to other brachytherapy techniques. We extended in text the phrase in line 91 as follows: “After a little training on the specific applicator, a physician confident with interstitial brachytherapy or surgery should be able to perform the treatment with ease”.

[C1.2a]: There is not enough high-quality evidence about the effectivity of radiotherapy in the management of cSCC.

[R1.2a]: We thank the Reviewer for comment that gives us the opportunity to highlight the role of radiotherapy.  In fact, the effectivity of radiotherapy in the management of cSCC is recognized by the NCCN guidelines that proposes radiotherapy as possible primary treatment in specific cases (We added references to NCCN guidelines in the introduction line 44 of the manuscript and to specific studies to support the effectivity of radiotherapy).

Anyway, we are aware of limits of radiotherapy with respect to surgery. This last allows a high quality control of surgical margins through PDEMA techniques that are not available for radiotherapy. This potential limitation has been discussed in the text in the discussion session as follows:

“Another limitation of this study is that we cannot provide complete lesion margins assessment with peripheral and deep en face margin assessment (PDEMA) technique but we have to rely on the general indications provided by NCCN and GEC-ESTRO guidelines for conventional radiotherapy in evaluating microscopic tumor invasion, target definition and treatment.”

[C1.2b] What clinical studies support that the alpha-particle administration procedure explained in the article is sufficient to obtain tumor coverage in cSCC?

[R1.2b] The only clinical study in man available now is the one cited in the paper by Popovtzer et al “Initial Safety and Tumor Control Results From a “First-in-Human” Multicenter Prospective Trial Evaluating a Novel Alpha-Emitting Radionuclide for the Treatment of Locally Advanced Recurrent Squamous Cell Carcinomas of the Skin and Head and Neck”.

The paper reports the results of the protocol that evaluates the safety and the tumour control rate of the alpha particle–based medical device DART. The alpha-particle administration technique remains the same and the aim of the present paper is to enhance the radiation delivery precision thanks to the employment of the template for seeds implantation. At the best of our knowledge, this is the major study regarding the alpha treatment of cSCC.

Due to the lack of studies about alpha particle treatments we followed the international guidelines for conventional brachytherapy to obtain tumor coverage, in particular employing margins (5-6mm) to define the Clinical Target Volume from the Gross Tumour Volume visible in CT.

In addition to this we specified in text in the methods section line 113: “At the end,  the tumour coverage is  ensured a-posteriori by either follow up in tumour complete response cases or with biopsy in suspected ones.” to explain further how tumour coverage has been evaluated

[C1.3]: How it is possible to detect microscopic subcutaneous expansions by the procedure proposed by the authors? Could the authors give enough data to support this statement?

[R1.4] We thank the Reviewer for pointing this out, but we didn’t intend to argue that with this procedure it is possible to detect microscopic subcutaneous expansions. The microscopic invasion is mentioned and considered only referring to the expansion from the CTV with respect the GTV,

May be there is a misunderstanding due to a confounding phrase in line 240:

 “by comparing the naked eye and PET/CT based lesion CTV, millimetric sized subcutaneous lesion expansions that are very difficult to see can be analysed”.

However we apologize with the Reviewer because the description on how we considered microscopic extension of the tumour was misleading. We also corrected a mistake found in the description of the procedure in the methods section 2.3.

 We modified the text in line 103 as follows:

“In the pre-treatment phase, the radiation oncologist draws the CTV directly on the patient's skin with a surgical pen and marks the needle insertion locations with a ruler, as shown in Figure. 2a considering 4-6 mm margins from the visible Gross Tumor Volume (GTV) to account for microscopic tumour invasion”(In the previous version of the manuscript 3 mm were mistakenly reported)

And in line 153 we added references employed to account for microscopic invasion:

“At this time, a PET/CT scan is performed to determine the extent of the tumor's subcutaneous extension, which is done by contouring the gross tumor volume (GTV) on the TPS and expanding it by 4-6 mm to define the CTV according to the NCCN and GEC-ESTRO guidelines [1,17]”

Furthermore we corrected figure 7A and 7B to correctly depict the expansion of the GTV of 5-6 mm instead of the 3 mm reported before. This was performed to match the correct dimension reported in figure 7B where the radio-opaque contour is at 5-6 mm from the visible GTV (confirmed by the dimension of the 2mm spaced template mesh)

[C1.4]: Although surgical excision is considered to be the primary treatment approach for curative treatment of cSCC, radiotherapy can play a role in both the definitive and adjuvant settings. Which results support that the alpha-particle administration is efficient in treating the real lesion borders in cSCC?

[R1.4]: We apologize with the Reviewer but the only clinical evidence for alpha particle treatment is provided by the clinical trial CTP-SCC-00 (https://clinicaltrials.gov/ct2/show/NCT03353077) and from it’s derived clinical trial CTP-SCC-02 (https://clinicaltrials.gov/ct2/show/NCT05125354) that is about to start recruiting. Unfortunately, evidence of treatment efficacy is only provided by complete response of the patient or biopsy for suspected recurrences and not with high quality histologic visualization of the entire marginal surface with techniques like peripheral and deep en face margin assessment (PDEMA).

However the demonstration of the efficacy of alpha-particle in treating the real lesion borders is beyond the scope of this paper and would require a dedicated clinical trial. We modified two misleading phrases as follows in line 235:

The direct visualization of the lesion on CT via the radio-opaque template will assist physicians and medical physicists in better evaluating lesion extension and, as a result, a better estimation of the number of sources required to achieve full tumour coverage while minimizing needle insertions, as shown in Figures 4A and 7B and C.

and in line 265:

 “New tumour contours and needle insertion locations coordinates are derived from the radio-opaque mesh and used to update the physical template, eliminating target misses and optimizing seeds positioning, as shown in Fig. 8D and Supplementary Fig. 2”.

Reviewer 2 Report

The paper gives a good account of the technique which is quite similar to that used in coventional brachytherapy

It is well written but would benefit from a review of English for minor grammatical errors and unusual words

All but one of the references are from related authors. This is hard to avoid in a new field but are they all necessary as the only people using this paper will already be familiar with the concepts?

The use of PET scans for superficial tumours is unusual, does it add much to CT alone?

The change between cm and mm as at the bottom of page 5 is confusing

I wonder if it is more suited to a technical note or as part of the results paper than as a full paper on its own. It is a a way of introducing more people to the concept of DaRT but previous papers do that

Author Response

First of all, we thank the Reviewer whose comments helped us to improve the manuscript.

Hereafter, “C# stands for Reviewer’s comments and “R#” for our replies. All the reference numbers (pages, citations, etc.) refer to the “new” version of the text, unless explicitly otherwise defined. In the answers, the references to manuscript’ sentences are reported between “< … >”. All changes to the text are reported in red or with Word Revision tool.

COMMENT OF REVIEWER#2:

[C2.1]: It is well written but would benefit from a review of English for minor grammatical errors and unusual words

[R2.1]: We thank the Reviewer for pointing this out, the manuscript is now revised for errors and unusual words

[C2.2]: All but one of the references are from related authors. This is hard to avoid in a new field but are they all necessary as the only people using this paper will already be familiar with the concepts?

[R2.2]: We thank the Reviewer for pointing this out. we tried to remove some unnecessary references. Furthermore, we added new references about brachytherapy and surgical guidelines for the treatment of cSCC

[C2.3]: The use of PET scans for superficial tumours is unusual, does it add much to CT alone?

[R2.3]: We agree with the Reviewer that PET scan is unusual for superficial tumours and CT is sufficient in most of the cases. PET can be informative only for tumours scarcely visible in CT such as planar lesions but always with the consulting of an expert nuclear physician for the correct uptake thresholding.

[C2.4]: The change between cm and mm as at the bottom of page 5 is confusing

[R2.4]:  We thank the Reviewer for the suggestion. We adjusted the text and figure 6 to uniform all the units to mm.

[C2.5]: I wonder if it is more suited to a technical note or as part of the results paper than as a full paper on its own. It is a way of introducing more people to the concept of DaRT but previous papers do that

[R2.5]:  We thank the reviewer for the comment but it is our opinion that the length of the paper is consistent for a full paper publication however, the editor will decide about this topic.

Round 2

Reviewer 2 Report

The authors have addressed the reviewer's comments as well as broadening the reference list. Minor English errors have been corrected and the paper reads well.